# Exploring Pharmacy Trainee Experiential Learning in a Full Scope Collaborative Rural Primary Care Practice: A Retrospective Chart Review

**DOI:** 10.3390/pharmacy9030155

**Published:** 2021-09-15

**Authors:** Sara Robinson, Feng Chang

**Affiliations:** School of Pharmacy, University of Waterloo, Kitchener, ON N2G 1C5, Canada; snrobinson@uwaterloo.ca

**Keywords:** pharmacy trainee, retrospective chart review, valuation, rural primary care

## Abstract

Despite reported benefits of pharmacy trainees (e.g., pharmacy students, pharmacy residents) in hospital settings, limited research on the impact of these trainees has been conducted in rural primary care. To explore the potential benefits and impact of pharmacy trainees practicing in a supervised collaborative rural primary care setting, a retrospective chart review was conducted. Drug therapy problems (DTPs) were classified using the Pharmaceutical Care Network Europe (PCNE V9) system. Valuation was measured using a validated tool developed by Overhage and Lukes (1999). Over 16 weeks on a part-time basis, pharmacy trainees (n = 3) identified 366 DTPs during 153 patient encounters. The most common causes for DTPs were related to patient transfers and the need for education. Drug level interventions carried out directly by trainees under supervision accounted for 13.1% of total interventions. Interventions that required prescriber authorization had an acceptance rate of 83.25%, 25% higher than previous acceptance rates found in urban primary care settings. About half (51%) of the interventions proposed and made by pharmacy trainees were classified as significant or very significant, suggesting these trainees added significant value to the pharmacy service provided to rural community residents. This study suggests that pharmacy trainees can be effective resources and contribute meaningfully to patient care in a collaborative rural primary care team setting.

## 1. Introduction

The scope of pharmacy practice has evolved in the last decade, giving pharmacists an integral role in medication management. In Ontario, pharmacists are now able to adapt prescriptions through altering the dose, formulation, regimen or route and have the ability to renew prescriptions for chronic conditions [1]. The expanded scope also allows pharmacists to administer certain vaccines, and administer other injectable or inhaled medications for patient education purposes [1]. With significant changes to the role of pharmacists, education provided to pharmacy trainees has also undergone restructuring. For example, transitioning from the Bachelors of Pharmacy to the Doctor of Pharmacy (PharmD) has occurred in many counties such as the United States, Benin, Hungary, Italy, Japan, South Korea, Pakistan, Saudi Arabia, Thailand, Republic of Congo and Nigeria [2]. As of 2020, all 10 pharmacy schools in Canada have adopted the Doctor of Pharmacy (PharmD) curriculum, in which pharmacy trainees are required to receive experiential learning in clinical settings in a second-entry undergraduate professional program setting [3]. In Ontario specifically, the University of Waterloo requires pharmacy trainees to complete three four-month cooperative work terms (work-based learning experience) and a six-month patient care rotation, while the University of Toronto requires trainees to complete 320 h of practice experiences and a nine-month patient care rotation [4,5]. Post-graduation, many graduates choose to continue their experiential training through one of the province’s 17 residency programs [6]. Only four of the residency programs offered in Ontario have an ambulatory care focus [7].

Active learning opportunities in clinical settings have been shown to improve critical thinking and clinical reasoning [8]. Recent graduates have reported feeling more prepared to provide full-scope services with ease, comfort and confidence [9]. The experiential component of the PharmD curriculum contributes not only to the overall competency of pharmacy trainees in applying pharmaceutical care, but it also benefits the clinical sites at which their learning took place [10,11,12,13]. In the hospital setting, pharmacy trainees have been shown to increase the accuracy of medication histories on admission, which in turn has been associated with reduced readmission rates [10]. A study completed at the Kings Country Hospital Center in New York, USA found that 30% of all interventions made by the pharmacy department were performed by pharmacy trainees, 92% of their interventions were accepted by the providers and 50% of the interventions were of a moderate to high significance level [11]. A similar study at the University Medical Center in Texas, USA found the most frequent interventions made by pharmacy trainees were recommendations for additional therapy, dose or frequency adjustments, drug monitoring and intravenous to oral therapy conversions, with an acceptance rate of 87% [11]. Both of these studies concluded that pharmacy trainees improved the quality of patient care and reduced the quantity of drug-related problems in the hospital setting [10,11]. Less research has been conducted on the role of pharmacy trainees in the primary care settings. When looking at pharmacy trainee interventions in a family medicine clinic, the main contributions included the provision of patient medication cards, education, medication reconciliation and monitoring recommendations [13]. Even though there was a lower acceptance rate of 58% when compared to the hospital studies (87–92%), pharmacy trainee interventions were still found to be impactful and resulted in significant cost savings [13].

Research is lacking for the role of pharmacy trainees in rural primary care. Geography is a major determinant of health, with those in rural communities having a higher incidence of chronic disease and mortality compared to those in urban locations [14]. This may be attributed to disproportionate access to health services [14]. Rural communities struggle to retain primary care providers and lack local diagnostic services and specialty practices in the area [14]. The Family Health Team (FHT) model of primary care has helped increase healthcare access to some regions. FHTs consist of multidisciplinary teams of healthcare providers, including physicians, nurse practioners, registered nurses, pharmacists, social workers and dieticians who work together to provide a wide range of services to their community [15]. FHTs are associated with comprehensive higher quality and cost-effective care [15]. With their increased scope of practice, pharmacy trainees may have the ability to fill in some gaps in healthcare in rural communities. This research was conducted to explore the potential benefits and impact of pharmacy trainees practicing in a supervised collaborative rural primary care setting.

## 2. Materials and Methods

The primary objective of this project was to critically review the identification of drug therapy problems (DTPs) and other interventions by pharmacy trainees from May to August 2019 at a rural FHT in Huron County, Ontario (Population: 59,297) [16]. The secondary outcomes of this study were to evaluate the value of service and the potential impact of pharmacy trainees’ recommendations on patient care.

In the context of our study, ‘pharmacy trainee’ is defined as: an individual engaging in the period of pre-registration training required under the Pharmacy Act including current university pharmacy students (e.g., PharmD student) or individuals who have already graduated from a university pharmacy level program and are in the process of post-graduate clinical training (e.g., pharmacy resident). During the study period, two third year PharmD students (out of four years) completed their 16-week co-operative academic term and one pharmacy resident (post-graduate year 1) completed a 6-week rotation at the site. The site includes 6 physicians, 2 nurse practitioners and 2 physician assistants with varying degree of prescribing authority, plus a team of clinicians including dietician, social worker and mental health workers. The site has a staff of one part-time pharmacist with full scope and medical directive for travel vaccine prescribing. The trainees were supervised by the pharmacist and provided direct patient care 1 to 2 days per week. Adult (18+) patients were able to schedule one-on-one appointments with the pharmacy through the FHT for specific medication related concerns by self-referral, through other team member referrals or through predefined criteria for programs conducted at the FHT (e.g., new prescription for anti-depressants, opioid therapy etc.). Trainees conducted patient work-ups beforehand using the FHT’s electronic medical records (EMR). During patient appointments, trainees identified DTPs, made care plans and provided suggestions and/or implemented interventions as appropriate. They also documented the encounter and provided follow-ups as determined by the care plan. Trainees engaged in discussions with the pharmacist and review of the care plan were held prior to, during and post appointment. These appointments were held with patients on days when the pharmacist was present. Care plan interventions were approved by the pharmacist prior to implementation for all trainees. Follow-up calls were carried out independently by the trainees based on the care plan and then reported to the pharmacist. Any new concerns identified by trainees during these follow-up calls were resolved by joint discussion with the pharmacist and revisions were made to the care plan if appropriate.

A retrospective chart review using pharmacy documentation and anonymization of data was conducted after all three trainees completed their academic terms. The following data were extracted and anonymized: appointment type, drug therapy problems (DTPs) identified, care plan, type of action (e.g., pharmacy implemented, primary care provider (PCP) authorized), actions taken, patient outcomes and time spent on encounter (direct and indirect). The DTPs and interventions were classified using the Pharmaceutical Care Network Europe (PCNE) V9.00 system [17]. PCNE’s classification for drug-related problem (DRPs) is a validated and internationally recognized classification system that includes five domains (problem/potential, cause, planned intervention, intervention acceptance and status of the DRP); commonly used to compare the results of a medication review, documentation and reimbursement in clinical practice [18,19]. These domains are further classified based on components, including but not limited to, treatment effectiveness (PCNE code P1), treatment safety (PCNE code P2), drug selection (PCNE code C1), drug form (PCNE code C2) and drug selection (PCNE code C3) [19]. Two PCNE domains (dispensing PCNE code C5 and drug use process PCNE code C6) were omitted as they were not relevant to this setting. The interventions were further classified into six categories of significance using an established valuation tool developed by Overhage and Lukes [20]. This valuation tool is used to evaluate the value of pharmacists’ clinical interventions while simultaneously measuring the severity of medication errors which can be used to develop corrective interventions and process improvements [20]. Using six categories (extremely significant, very significant, significant, somewhat significant, no significance and adverse significance), the potential impact of the pharmacists’ recommendation on patient care is assessed (value of service) [20]. The inappropriateness of the order, or its deviation from the standard of practice, otherwise known as the severity of error, in this tool is classified as potentially lethal, serious, significant, minor or no error [20]. Data were categorized by one team member with an additional reviewer for verification. Any conflicts were resolved by discussion. This study received ethics clearance through the University of Waterloo (ORE # 41414). 

## 3. Results

### 3.1. DTPs Identified and Interventions Made by Pharmacy Trainees

Classification of appointment type was specified at the time of booking (e.g., ‘polypharmacy’, ‘medication review’) and time allocated for direct care at the time of booking was standardized to 1 h to allow for complex cases. The average time spent per appointment was patient and case dependent. The indirect time spent per appointment refers to the amount of time needed for prior work-up, documentation and follow-up. A significant portion (47%) of appointments were opioid reviews, in which patients on opioid therapy received education on effectiveness and adverse effects, signs of overdose and actions to address an overdose (Table 1).

A total of 366 DTPs were identified during 153 patient encounters which resulted in 133 EMR updates for all 133 patients. Overall every patient had at least one discrepancy in their medication list on the EMR chart. The majority of DTPs (74%) were classified under treatment safety or medication reconciliation (Table 2). Treatment safety was classified by an adverse drug effect occurring, or having the possibility to occur, which included dosages that were too high and durations that were too long.

The causes of the identified DTPs varied with patient transfer related DTPs being the most common (Table 3). In these cases, the EMR medication list was not current and medication reconciliation was required. Approximately 22% of DTPs were caused by poor understanding of medication instructions requiring further education. Surprisingly, drug selection, specifically “no or incomplete drug treatment in spite of existing indication”, attributed to 22% of the identified DTPs.

Acceptance of proposed intervention was based on chart documentation (Table 4). Examples of documentation for ‘accepted but not implemented’ included: prescriber suggested but patient either did not implement on follow-up or it was declined at the time of discussion. A total of 72 interventions required PCP authorization and a significant amount of these interventions were accepted by the prescriber. It should be noted that not all patients seen required follow up by the pharmacy and not all patients had documented outcomes.

The corresponding interventions to DTPs were mostly made at the prescriber and patient level (Figure 1). Drug level interventions, such as dose alterations and alternative therapies, made directly by the trainee constituted 13% of all cases. Medications that were discontinued included inappropriate medications (e.g., diphenhydramine, dimenhydrinate) and unnecessary medications (e.g., long-term proton pump inhibitor for asymptomatic GERD). Examples of dose adjustments included lowering dosages of antihypertensive medications for symptomatic hypotension and bradycardia, and increasing dosages of antidepressants for depression and neuropathic pain to reach target dosing range. Initiating new therapy were also among the recommended interventions and included prescribing travel vaccinations and recommending over-the-counter products such as B12 supplements.

### 3.2. Valuation of Pharmacy Trainee Interventions

Using Overhage and Lukes’ (1999) valuation tool, one fifth of the interventions were determined to be very significant (Figure 2). This included interventions that resolved adverse effects (e.g., reducing hydrochlorothiazide dose to resolve hypotension), optimized response (such as increasing venlafaxine dose for depression control) and adding therapy for untreated indications (e.g., prescribing travel vaccinations). Approximately a third of the interventions were categorized as significant, signified by improving patient care (e.g., opioid education and contract signing), incorporating cost and/or convenience measures (e.g., recommending combination pill to reduce pill burden). Improvements in vitals was the most commonly noted patient outcome followed by resolution of ADR such as dizziness, fatigue and diarrhea (Table 5). Almost half of interventions deemed somewhat significant consisted mostly of medication reconciliations, for which the significance of having an updated medication list varied.

## 4. Discussion

The majority of trainee interventions were found to be significant in improving medication or treatment safety. Medication reconciliation and patient education were the most common interventions made by pharmacy trainees in this study. Every patient had at least one discrepancy in their medication list on the EMR chart. The pharmacy trainees updated this for each patient, allowing other allied health professionals to have access to an accurate medication history. The frequency of patient-focused interventions were similar to previous findings in primary care, which saw patient counseling, providing medication cards and medication reconciliation as the most frequent interventions made by pharmacy trainees [13]. Comparatively, previous pharmacy trainee interventions conducted in hospital settings were found to be more medication specific, with “drug information” and “recommending alternative agents” most frequent in one study, and “additional therapy needed” and “dosage too low” most frequent in another [11,12].

In primary care, pharmacy trainees are able to deliver patient-focused care, with a noted emphasis on patient counseling and education.

Unique to our study, pharmacy trainees were found to make 48 interventions at the drug level independently, under a pharmacist’s supervision but without requiring the PCP’s approval. This contributed to 13% of the total interventions made. The three most frequent drug interventions were discontinuing a medication, adjusting the dose of a medication and starting new therapy. These pharmacy trainee-led interventions corresponded to direct patient outcomes. For example, during documented follow-up, increasing dosages of antidepressants were associated with decreased scores on the PHQ-9, a questionnaire used to monitor depression severity and response to therapy; decreasing antihypertensive medication dosages improved patients’ blood pressure and heart rate to target ranges [9,10,11,12]. Previous research on pharmacy trainee contributions in hospital and primary settings required the PCP to accept the recommendation and intervene for all actions at the drug level [10,11,12,13]. This is attributed to the expanded scope of pharmacy practice in Ontario, in which pharmacists are able to alter the dose, formulation, regimen and route of medications independently. This can reduce the need for separate patient appointments with the PCP and lead to improved system efficiency.

Pharmacy trainee interventions that required PCP authorization had an acceptance rate of 83.3%. This was a higher rate than previous research in primary care settings, which found an acceptance rate of 58% [13]. However, the acceptance rate in hospital settings was higher in previous research at 87–92% [11,12]. Infrastructure differences between study sites may have contributed to these contradictory findings. In our primary care setting, interventions were proposed via EMR messages to prescribers. These messages may have been missed or archived. Comparatively, in hospital settings, pharmacy trainees often directly communicated with prescribers during patient rounds or phone calls [11,12]. Patient factors and values may have also played a role in the resulting acceptance rate. For example, seven proposed interventions were accepted by the PCP, but were declined by the patient when the PCP tried to initiate the change in therapy later on. For the 12 interventions not accepted by the PCP, the study was not able to capture whether these were refusals, or if the PCP would reassess the intervention in the future. Further research can be carried out to identify barriers and strategies for implementation of pharmacy trainee interventions.

Over 50% of the interventions proposed and made by pharmacy trainees were determined to be significant to very significant, using the valuation tool by Overhage and Lukes [17]. The institutional study at University Medical Center used a similar tool to assess the significance of their pharmacy trainee interventions, in which three categories of clinical importance were analogous to Overhage and Lukes’ categories of significant, very significant and extremely significant [12,17,21]. Researchers at the University Medical Center found that 36.4% of the pharmacy trainee interventions were of minor (significant), 56.1% were of moderate (very significant) and 7.5% were of severe clinical importance (extremely significant) [12]. Patients admitted to the hospital tend to be more acutely ill and in greater critical condition than those in scheduled appointments under primary care. As such, the hospital setting may inherently contain opportunities for pharmacy trainees to propose interventions with higher clinical importance.

This study had a number of limitations. To begin, the findings were based on a small sample size of pharmacy trainees and a short study period of 4 months, although representative of typical experiential work-terms. Working in a part-time setting could also have led to delays in patient feedback. In this regard, the study site is representative of the region with the resources invested in its multidisciplinary team members, the patient population drawing from several neighboring rural townships and types of clinical issues encountered. Data collection was completed through a retrospective chart review of the pharmacy trainees’ documentation, and as such, there is a possibility of reporting bias. This risk was mitigated by the staff pharmacist reviewing the documentation after each appointment for accuracy. In addition, this study did not have the resources to convene an expert panel to perform the valuation analysis which could potentially lead to the significance being under or overestimated. To reduce the risk of error, the valuation was completed by two individuals. Limited follow up data is another limitation of this study as not all patients seen required follow up by the pharmacy and not all patients had documented outcomes. A number of factors may have contributed to this finding. For example, follow ups provided by other clinicians, especially external specialists, were not always explicated documented. Depending on the nature of the issue, the patient may not have completed follow up at the time of data extraction. This was also compounded by the fact that the patient could also have been difficult to reach or the documentation in the chart was vague or unclear for the related outcome. This is reflective of the real-world documentation challenges that exist, despite progresses made in electronic patient records.

To our knowledge, this was the first study completed on the role of pharmacy trainees in a collaborative rural Ontario primary care setting, and one of the few to be completed in a primary care setting. The benefits and positive impacts captured during this study could contribute to the retention of primary care providers and pharmacists in rural community settings by: providing needed support to current rural clinicians and community pharmacies and; through experiential learning opportunities, provide pharmacy trainees the opportunity to consider practicing in these regions. Other counties, such as Australia and the United States, have created rural immersion experiential learning opportunities for their pharmacy trainees to help foster an appreciation for the expanded scope of practice many rural pharmacists experience with reported trainee benefits including significant increases in understanding and knowledge about rural health, relationship-building in the community, a better understanding of social and cultural dynamics of remote location practice settings and underserved populations, more time with faculty/supervisors, and interprofessional collaboration experience [22,23,24,25,26].

Pharmacy trainees were found to be valuable members of the collaborative team with the ability to make a variety of clinical interventions. Uniquely, this study showed that pharmacy trainees are able to make drug level interventions independently of the PCP, which may confer healthcare savings and result in patient health benefits. Despite the promising results found in this study, the study’s scope was limited as it only contained results based on one site. It may be beneficial to compare different experiential learning programs available across Ontario or national pharmacy schools to better determine the value pharmacy trainees add to primary care practice, particularly in rural and underserviced practice sites. It may also be of value to explore differences in experiential models, learner environments and other logistical factors that could affect the learning experience and value provided.

## 5. Conclusions

Pharmacy trainees were valuable patient care resources and their clinical recommendations were well-accepted at a full scope collaborative rural primary care practice. Further research to delineate the clinical and financial impact of trainee work will help to optimize the experiential model with lessons transferable to other jurisdictions.

## Figures and Tables

**Figure 1 pharmacy-09-00155-f001:**
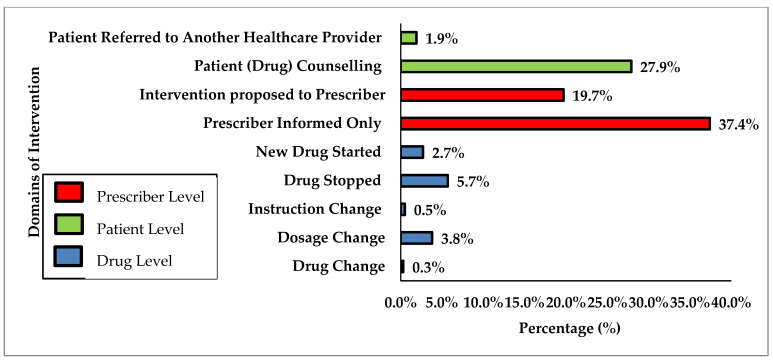
Classification of Interventions (n = 366).

**Figure 2 pharmacy-09-00155-f002:**
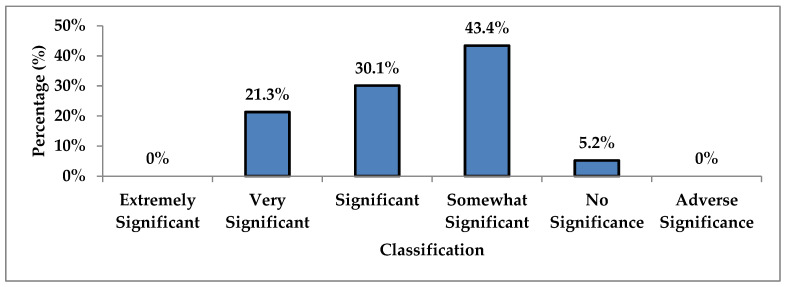
Valuation of Interventions (n = 366).

**Table 1 pharmacy-09-00155-t001:** Categorical Data of Patient Appointments seen by Pharmacy Trainees.

Characteristic	n (%)
Number of appointments	153
Polypharmacy	54 (35)
Opioid review	72 (47)
Travel clinic	2 (1)
Referral	5 (3)
Follow-up	20 (13)
Average time spent per appointment (min)	72
Direct contact time spent with the patient/caregivers	40.5 (56)
Indirect	31.5 (44)

**Table 2 pharmacy-09-00155-t002:** Classification of DTPs Identified (n = 366) by Pharmacy Trainees.

PCNE Code	Domain and Classification	n (%)
	Treatment Effectiveness	76 (20.7)
P1.1	No effect of drug treatment/therapy failure	14
P1.2	Effect of drug not optimal	38
P1.3	Untreated symptoms or indication	24
	Treatment Safety	139 (38.0)
P2.1	Adverse drug event occurring (or possibility to occur)	139
	Other	151 (41.3)
P3.1	Problem with cost-effectiveness of treatment	3
P3.2	Unnecessary drug treatment	15
P3.3	Other (specify; medication reconciliation required)	133

**Table 3 pharmacy-09-00155-t003:** Classification of causes of DTPs identified (n = 366).

PCNE Code	Domain and Classification	n (%)
	Drug Selection	81 (22.1)
C1.1	Inappropriate drug according to guidelines	13
C1.3	No indication for drug	9
C1.4	Inappropriate combination of drugs or drugs and herbals	13
C1.5	Inappropriate duplication of therapeutic group/ingredient	5
C1.6	No or incomplete drug treatment in spite of existing indication	41
	Dose Selection	33 (9.0)
C3.1	Drug dose too low	11
C3.2	Drug dose too high	20
C3.3	Dosage regimen not frequent enough	1
C3.4	Dosage regimen too frequent	1
	Treatment Duration	7 (2.0)
C4.1	Duration of treatment too short	1
C4.2	Duration of treatment too long	6
	Patient Related	90 (24.6)
C7.1	Uses/takes less drug than prescribed or does not take	9
C7.7	Inappropriate timing or dosing intervals	2
C7.10	Unable to understand instructions properly (requires education)	79
	Patient Transfer Related	133 (36.3)
C8.2	No updated medication list available	133
	Other	22 (6.0)
C9.1	No or inappropriate outcome monitoring	9
C9.2	Other (specify; prescriber requires drug information, patient needs referral to another healthcare professional)	13

**Table 4 pharmacy-09-00155-t004:** Classification of interventions proposed to PCPs (n = 72).

PCNE Code	Domain and Classification	n (%)
	Intervention Accepted	60 (83.3)
A1.1	Intervention accepted, fully implemented	53
A1.3	Intervention accepted, but not implemented (patient declined)	7
A2.2	Intervention not Accepted	12 (16.7)
	Intervention not accepted: no agreement	12

**Table 5 pharmacy-09-00155-t005:** Documented patient outcomes (n = 26).

Outcome	n
Reduction in Patient Health Questionnaire (PHQ-9) Score (depression control)	4
Improvement in vitals (heart rate, blood pressure)	10
Improvement in blood work (hemoglobin, B12, potassium)	3
Resolution of ADR (dizziness, fatigue, diarrhea)	5
Reduced pill burden (with no rebound symptoms)	4

## Data Availability

Data sharing not applicable.

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
