# Peer review of "Exploring Pharmacy Trainee Experiential Learning in a Full Scope Collaborative Rural Primary Care Practice: A Retrospective Chart Review"

_pharmacy, 2021, doi:10.3390/pharmacy9030155_

Round 1

Reviewer 1 Report

This manuscript describes and analyses the Drug Therapy Problems, handled by three pharmacy trainees (2 undergraduate, 1 postgraduate) in a rural pharmacy in Huron, Ontario, Canada. The manuscript is interesting and describes the situation succinctly. However, a few changes can improve readability.

Major remarks:

  1. The presentation of the Results is somewhat confusing, and can be better structured. The main findings are presented in table format and need not to be repeated in the text. I suggest that the text of the Result-section is written as a ‘reading guide’, where the authors draw attention to certain aspects of the quantitative findings, which they find remarkable or interesting. All tables need to be referred to in the text, whenever relevant (at the moment no reference is made in the text to tables 3-6!). Furthermore, table 4 and fig.1 contain the same information; this is unnecessary.
  2. Related to the previous point is the observation that new results are mentioned in the Discussion section (e.g. lines 170-173, lines 186-189 and lines 205-208). It would be better if these results are mentioned in the Results section, and that the authors opinion or evaluation of the finding is discussed in the Discussion section.
  3. It is unclear whether interventions by the student trainees need approvement by the supervising pharmacist. It is indicated that certain interventions need to be submitted tot the prescriber for approval, but I can imagine that also interventions, which are legally the responsibility of a pharmacist, cannot be exercised by a trainee independently. Please indicate in the manuscript how this was handled in this case. Is there a difference between handling of interventions between undergraduate and postgraduate trainees?
  4. A major limitation of this study appears to be the fact that the study was done in one pharmacy location. I know that this limitation is mentioned in the text (lines 239-244) but it would be helpful if the authors discussed whether this pharmacy can be considered representative for pharmacies in rural areas of Ontario, Canada. Furthermore, how is the role of the supervising pharmacist evaluated?

Minor remarks:

  1. Line 32. I don’t understand the limitation “for educational purposes”.
  2. Line 66. Cost savings of $ 61,855 don’t mean anything if no reference is made to the total costs.
  3. Line 92. Supervision was exercised by a part-time pharmacist. Therefore, patient-feedback can be delayed, depending to the working hours/days of the pharmacist (see also remark 3, above). Please clarify.
  4. Table 1. Average time spent is separated in Direct and Indirect time. ‘Direct time’ probably represents contact time with patients, but this needs to be specified.

9. Line 283. Ref. 9. ‘P.T.’ = unknown abbreviation. 

Reviewer 2 Report

1. The manuscript focuses on the Canadian perspective. Seeing that the manuscript is intended for an international audience, it is suggested that in the Introduction and Discussion section, reference to works from other academic experiences in other regions is invovked.

2. In the Method section there is lack of clarity of how the patients were sampled to be called for the trainee-review

3.  The Results section requires improvement:

- the section starts off with a table (Table 1) with very little explanation

-tables are included within the section, incorrectly referred to in the text (eg Page 4 line 126 reference to Table 2) making the section  confusing and not transposing clarity of data presented. 

-lack of clarity with regards to how many patients were reviewed, average number of medications per patient, were they receiving chronic medications, conditions captured

Author Response

Please see the attachment, thank you!

Reviewer 3 Report

Thank you for the opportunity to review this interesting paper on the role of learners in a rural primary care practice. The paper is generally well written but I have some suggestions, many of which relate to understanding local terminology and context. For an international audience there are some aspects needing additional explanation. 

Throughout the paper the use of the work trainee is inconsistent and has not been defined. I take it to mean a combination of the students and the resident however sometimes (eg section 3.1) students are specifically referred to. Please define what this term means in the context of your paper, and ensure the terminology is applied consistently throughout

Introduction

  • line 37: please define what is meant by a "cooperative work term". I am aware of this term but I think it may be specific to Canada and many readers will not understand what it refers to

Materials and Methods

  • line 87: the students were in third year--can you please explain whether this is the final year of their PharmD ie are they about to graduate/register or are they mid way through their degree?
  • line 95: patients were able to schedule one on one appointments--who with? Did they specifically book an appointment with a pharmacist (or "trainee") or did they book an appointment with a generic health care professional in the team?
  • line 99 the trainees discussed with the pharmacist throughout the appointment but the pharmacist only worked part time. Who played this role when the pharmacist was not on duty? Did the "trainee" see patients independently or was the pharmacist present throughout the appointment?
  • line 108: for readers unfamiliar with the PCNE system and the validation tool [ref 17] I wonder whether having some more information (or an appendix) would  be helpful to the reader to better understand how the classifications were done?

Results

  • Table 1: was the type of appointment specified at the point of booking or was this classification decided/defined later on? How was the time allocation for direct and indirect decided?
  • Table 4: 137/366 interventions were "prescriber informed only" This may need some further investigation in the discussion section. What sort of things fall into this category? If they are for information only are they really interventions? I would imagine this is covered in the original reference [16] but I think worth of some words in this paper. 
  • does Figure 1 add anything to table 4? One of them could potentially be deleted
  • Table 6: the patient follow up data is not mentioned in the methodology. Were all patients followed up and only 26 had documented outcomes? Or were only 26 followed up? This probably needs an addition to the methodology to explain the results here.

Discussion

  • line 165: should "pharmacy" interventions be "trainee" or "student" interventions?
  • line 167 "Every patient had a least one discrepancy in their medication list on the EMR chart." I don't think this data is presented in the results?
  • line 175: the sentence about pharmacy student interventions in the hospital setting could be reworded to make it clearer it is referring to previously published work not related to this study
  • line 200: if interventions were proposed via EMR messages to prescribers, how was the data in table 5 determined? How did you know what was accepted but not implemented vs not accepted? I can see how you would know which ones were accepted and implemented from chart review
  • I would like to see the discussion loop back to the introductory remarks about the rural health workforce. How can these findings assist  in meeting these needs? Is there a need (for example) to amplify experiential offerings in these settings as a mechanism to support the workforce given the results of your work?

Conclusions

  • line 247 typo in "valuable"

References

  • date of access for online references is now quite old and may need to be updated

Abstract

  • I think this could be improved to better focus on the important sections of the paper. For example the opening sentence about increased experiential training, in particular at the post graduate level, are not really explored in detail and are not the focus of the paper
